# Psychological Characteristics and Quality of Life of Patients with Upper and Lower Functional Gastrointestinal Disorders

**DOI:** 10.3390/jcm12010124

**Published:** 2022-12-23

**Authors:** Seung-Ho Jang, Suck-Chei Choi, Yong-Sung Kim, Han-Seung Ryu, Sang-Yeol Lee, Won-Myong Bahk

**Affiliations:** 1Department of Psychiatry, School of Medicine, Wonkwang University, Iksan 54538, Republic of Korea; 2Department of Internal Medicine, School of Medicine, Wonkwang University, Iksan 54538, Republic of Korea; 3Digestive Disease Research Institute, School of Medicine, Wonkwang University, Iksan 54538, Republic of Korea; 4Department of Psychiatry, College of Medicine, The Catholic University of Korea, Seoul 07345, Republic of Korea

**Keywords:** functional gastrointestinal disorders, gut–brain axis, psychological intervention, quality of life

## Abstract

Background: This study aimed to identify the differences in the psychological characteristics of the anatomical location of functional gastrointestinal disorders (FGIDs) and the factors that influence the quality of life (QOL). Methods: Altogether, 233 patients with FGIDs were classified into the upper gastrointestinal disorder (UGID; *n* = 175) group and the lower gastrointestinal disorder group (LGID; *n* = 58). Psychological characteristics and QOL were evaluated using the validated questionnaires. Results: The LGID group demonstrated higher scores in ‘emotional depression’ than the UGID group in depressive symptoms (*t* = −3.031, *p* < 0.01). A significant difference was observed between groups in ‘significant others’ in social supports (*t* = 2.254, *p* < 0.05). Significant differences were observed between the groups in hardiness (*t* = 2.259, *p* < 0.05) and persistence (*t* = 2.526, *p* < 0.05) in resilience, while the LGID group demonstrated significantly lower scores than the UGID group in ‘negative affectivity’ in type-D personality (*t* = −1.997, *p* < 0.05). Additionally, the LGID group demonstrated lower QOL than the UGID group (*t* = 2.615, *p* < 0.05). The stepwise regression analysis on QOL involved depression, resilience, social support, and childhood trauma, which accounted for 48.4% of the total QOL explanatory variance. Conclusions: Psychological characteristics and QOL significantly differed when FGIDs were classified according to anatomical location. Thus, psychological interventions customized for each type of FGIDs may be necessary for effective treatment.

## 1. Introduction

Despite chronic gastrointestinal (GI) symptoms, challenges remain in the diagnosis and effective treatment of functional gastrointestinal disorders (FGIDs) due to the lack of objective indicators arising from the absence of organic abnormalities. For this reason, the Rome criteria were proposed as a guideline for the diagnosis and treatment in research and clinical practice on FGIDs [1].

Traditionally, FGID has been categorized by anatomic regions such as esophageal, gastroduodenal, bowel, gallbladder, and anorectal disorders. Local factors such as motility disorder, altered immune function, and gut dysbiosis (i.e., no “healthy” microbiota) are some of the main underlying pathophysiologies [1], however, central factors such as altered central processing and psychological disturbance can also play important roles. The symptoms of FGIDs are generated by complex interactions between the aforementioned local and central factors. Therefore, FGIDs have been understood from a biopsychosocial perspective and have been re-defined as “disorders of the gut–brain interaction” according to the Rome IV criteria [2].

According to studies on the gut–brain axis, altered cognition or emotions due to environmental stressors, fear, or anger causes GI symptoms through descending cortico-limbic signals [3].

The bi-directional interactions of gut–brain axis in FGIDs have also been demonstrated in epidemiological studies [4]. FGIDs are closely associated with psychiatric symptoms. Using meta-analysis, we previously reported that depression and anxiety levels are higher in patients with irritable bowel syndrome (IBS) patients as compared with healthy controls [5]. Halland et al., have reported that IBS symptoms were significantly correlated with histories of childhood sexual, emotional, or verbal abuse, proposing that childhood psychosocial trauma can impair resilience and the ability to recover from chronic stress, causing various physical symptoms that cannot be organically explained [6]. Psychological disorders also are closely associated with FGIDs. A study of patients with dysthymia has revealed that 25% met the criteria for IBS, while 39% of patients with affective disorders met the criteria for IBS [7]. In some patients with IBS, psychiatric disorders precede IBS onset, and these patients may respond better to psychiatric treatment [8].

Quality of life (QOL) is an individual’s perception of their status in life in relation to their goals, expectations, norms, and interests, within the context of the culture and value system in which they are situated [9]. Quality or lack of social support is closely related to IBS [10] and is an essential factor in overcoming IBS [11]. In particular, appropriate social support reduces the severity of IBS by lowering the level of stress experienced [12]. Social support appears to work by boosting the immune system, especially among people who are experiencing stress [13]. QOL can be a major factor that influences symptoms, in addition to serving as the ultimate goal for patients undergoing treatment. Kim et al. have reported that the QOL of patients with FGIDs were closely correlated to resilience and other various psychiatric symptoms [14].

Somatization is defined as “a tendency to experience and communicate somatic symptoms that are unaccounted for by pathological findings in response to psychosocial stress and to seek medical help for them” [15]. Further, somatization is associated with gastric sensitivity, gastric emptying, and symptom severity in functional dyspepsia (FD) [16], and it affects extraintestinal symptoms in IBS [17]. Since the relationship between type D personality (a tendency towards both negative affectivity and social inhibition) and somatization has recently been reported [18], research on the relationship between type D personality and GI symptoms is warranted. 

The esophagus, stomach, and colon have different distributions of the autonomic nervous system and differences in terms of nervous system involvement in symptom development [19]. Therefore, it can be hypothesized that mental symptoms in UGID and LGID may differ, ultimately resulting in a difference in QOL between patients with UGID and LGID due to different effects of the brain on the esophagus, stomach, and colon.

Thus, this study aimed to identify the differences in psychological characteristics between upper and lower functional GI disorders and factors influencing the QOL of patients with FGIDs.

## 2. Materials and Methods

### 2.1. Participants

This study was conducted from March 2020 to December 2021 and included 233 patients who underwent gastrointestinal endoscopy, 24 h pH-impedance test, and esophageal manometry test in a gut–brain mental health clinic at a university hospital and were diagnosed with FGIDs by a gastroenterologist. The participants were classified into globus, functional heartburn (FH), FD, and IBS, then grouped into the upper GI disorders group (UGID, *n* = 175, globus; 31, FH; 50, FD; 94) and lower GI disorders group (LGID, *n* = 58, IBS; 58) according to the innervation of parasympathetic nerves (vagus nerve: UGID, sacral nerve: LGID) [19]. The inclusion criteria were as follows: (i) those who understand the purpose of the study and consent to participate; (ii) those aged 18–75 years; and (iii) those who were tested for upper endoscopy, colonoscopy, abdominal imaging, 24 h pH-impedance test, and esophageal manometry test if they had a clinical symptom. The exclusion criteria were as follows: (i) those with a history of GI surgery except appendectomy; (ii) those with inflammatory bowel disease, malignancy, or a systemic disease requiring drug treatment; (iii) those who were pregnant or lactating; (iv) those with a hepato-biliary disease; and (v) those with a history of mental disorder. Each participant underwent a psychiatric assessment for symptoms and a psychiatric interview. Informed consent was obtained from all the participants. This study was approved by the institutional review board (IRB) of Wonkwang University Hospital (IRB approval number: WKUH 2018-04-010).

### 2.2. Diagnostic Questionnaire of Functional Gastrointestinal Disorders 

The Korean Rome III criteria-based questionnaire was administered to the study subjects [20]. FGIDs were diagnosed when no abnormality was found in objective investigations that were utilized to explain the symoptoms in patients with the following GI symptoms that satisfied the Rome III criteria. These investigations included 24 h pH-impedance test, manometry, esophagogastroduodenoscopy, colonoscopy, radiology, and laboratory study. Globus was diagnosed when patients complained of a persistent or intermittent, sensation of a lump or foreign body in the throat that was not painful. FH was diagnosed when patients complained of burning retrosternal discomfort or pain [21]. FD was diagnosed when patients had one or more symptoms of postprandial fullness, early satiation, epigastric pain, and epigastric burning [22]. IBS was diagnosed when patients complained of recurrent abdominal pain or discomfort associated with relief from defecation. Its onset is associated with a change in frequency or form of stool [23].

### 2.3. Korean Version of the Beck Depression Inventory 2nd Ed. (K-BDI-II)

Depressive symptoms were evaluated using the K-BDI-II, which is a self-reported depression scale comprising 21 questions, for which the participant is asked to select the most suitable response out of four statements. It comprises sub-domains of emotional, cognitive and somatic. Each question is scored out of three points (0–3), with the total score ranging from 0 to 63. At a cut-off of 10 points, the participants could be grouped into the depressed group and normal group, where a score of 10–15 points is categorized as mild, 16–23 points is categorized as moderate, and 24–63 points is categorized as severe [24]. The K-BDI-II was translated into Korean by Sung et al. and evaluated for validity and reliability [25].

### 2.4. Korean Version of the Beck Anxiety Inventory (K-BAI)

The K-BAI was used to measure anxiety. It comprises sub-domains of reflecting subjective, neurophysiological, autonomic, and panic symptoms of anxiety, measuring the level of anxiety experienced during a week on a 4-point scale. The total score ranges from 0 to 63, with a score of ≥22 being classified as a high-risk group among adults. Translation into Korean and the assessment of reliability were performed by Yook et al. [26].

### 2.5. Korean Version of the Childhood Trauma Questionnaire (CTQ-K)

Childhood trauma was assessed using the CTQ-K, which is a self-reported assessment tool that measures childhood trauma and comprises five sub-domains of emotional neglect, physical neglect, emotional abuse, physical abuse, and sexual abuse. Translation into Korean and the assessment of reliability were performed by Kim et al. [27].

### 2.6. Multi-Dimensional Scale of Perceived Social Support (MSPSS)

The MSPSS developed by Zimet et al. was used to assess the level of social support [28]. The MSPSS consists of three sub-domains comprising four questions of ‘family’, four questions on ‘friend’, and four questions on ‘significant other’. Each question is measured on a scale of 1–5 points, with higher points indicating greater levels of social support. Translation into Korean and the assessment of reliability were performed by Shin et al. [29].

### 2.7. Korean Version of the Connor–Davidson Resilience Scale (K-CD-RISC)

Resilience was assessed using the K-CD-RISC developed by Connor and Davidson [30]. The K-CD-RISC comprises five sub-domains of hardiness, persistence, optimism, support, and spiritual influence, which are measured on a 5-point scale with higher scores indicating greater resilience. Translation into Korean and the assessment of reliability were performed by Beak et al. [31].

### 2.8. Korean Version of the Type-D Personality Scale-14 (K-DS-14)

Type-D personality was assessed using the 14-item K-DS-14, which includes the sub-domains of negative affectivity (NA) and social inhibition (SI), which are each measured in a 5-point scale. Individuals that have a score of ≥10 in both NA and SI are identified as having the type-D personality. Translation into Korean and standardization were performed by Lee et al. [32].

### 2.9. World Health Organization (Geneva) Quality of Life—Brief Version (WHOQOL-BREF)

To measure the QOL, this study used the WHOQOL-BREF, an abbreviated QOL questionnaire developed by the World Health Organization Quality of Life Group. It comprises 26 items including the following: two items on the overall QOL, six items on psychological domain, seven items on the physical domain, eight items on environmental domain, and three items on social relationship domain. Each item is evaluated on a scale ranging from 0 to 5 points with a score of 60 points as the optimal cut-off score, and higher scores indicate a higher QOL. Min et al. translated the questionnaire into Korean and verified its validity and reliability [33].

## 3. Statistical Analysis

Demographic data and psychological characteristics were compared, from which the mean and standard deviations for continuous variables and frequency and proportion of categorical variables were presented. An independent t-test was performed in order to compare the differences between groups, while a Pearson correlation test was performed to analyze the correlation between QOL and psychological characteristics. A stepwise regression analysis was performed to identify the factors influencing the QOL of FGIDs patients. All statistical tests were two-tailed. *p* < 0.05 was considered statistically significant. The collected data were analyzed using the Statistical Package for the Social Sciences (SPSS, Version 21, IBM, Chicago, IL, USA).

## 4. Results

### 4.1. Demographic and Clinical Characteristics of Participants

No significant differences in sex, age, marital status, education, income, smoking, alcohol consumption, use of antiacid agents, and chronic disease were observed between the UGID and LGID groups (Table 1).

### 4.2. Comparison of Depression and Anxiety between the UGID and LGID Groups

A significant difference was observed in the sum of depressive symptoms between the UGID (18.53 ± 10.68) and LGID (23.10 ± 12.68) groups, as well as in the sub-domain of ‘emotional’ (UGID (7.97 ± 5.49), LGID (10.64 ± 6.67); (*t* = −3.031, *p* < 0.01)). No significant difference in anxiety was observed between both groups (Table 2).

### 4.3. Comparison of Social Support and Childhood Trauma between the UGID and LGID Groups

A significant difference among the subdomains of social support between groups was only observed in ‘significant others’ (UGID (11.23 ± 3.83) vs. LGID (9.93 ± 3.70)) (*t* = 2.254, *p* < 0.05). This suggests that the UGID group experienced greater social support from significant others compared to the LGID group. In childhood trauma, no significant difference was noted between both groups (Table 3).

### 4.4. Comparison of Resilience and Type-D Personality between the UGID and LGID Groups

In the sum of resilience, a significant difference was observed between UGID (59.19 ± 17.78) and LGID (53.21 ± 18.51) (*t* = 2.198, *p* < 0.05) groups, as well as in the sub-domains of hardiness [UGID (20.72 ± 6.91) vs. LGID (18.36 ± 6.83); *t* = 2.259, *p* < 0.05] and persistence [UGID (19.89 ± 6.85) vs. LGID (17.19 ± 7.40); *t* = 2.526, *p* < 0.05]. In type-D personality, a significant difference in NA was observed between groups (UGID (11.08 ± 6.53) vs. LGID (13.07 ± 6.71); *t* = −1.997, *p* < 0.05) (Table 4).

### 4.5. Comparison of QOL between the UGID and LGID Groups

A significant difference in the sum of QOL was observed between the UGID (78.78 ± 12.84) and LGID (73.66 ± 13.19) groups (*t* = 2.615, *p* < 0.05). Among the sub-domains, significant differences between groups were observed in overall wellbeing [UGID (5.55 ± 1.50) vs. LGID (4.74 ± 1.59); *t* = 3.515, *p* < 0.01], physical [UGID (20.15 ± 3.91) vs. LGID (18.67 ± 4.12); *t* = 2.471, *p* < 0.05], psychological [UGID (18.61 ± 3.21) vs. LGID (17.48 ± 3.77); *t* = 2.207, *p* < 0.05], and social sub-domains [UGID (8.50 ± 1.96) vs. LGID (7.71 ± 2.17); *t* = 2.613, *p* < 0.05] (Table 5).

### 4.6. Bivariate Associations between QOL and Psychological Variables in Patients with FGIDs

The QOL demonstrated a positive correlation with social support (r = 0.411, *p* < 0.01) and resilience (r = 0.488, *p* < 0.01), and a negative correlation with depressive symptoms (r = −0.610, *p* < 0.01), anxiety (r = −0.473, *p* < 0.01), and type-D personality (r = −0.491, *p* < 0.01) (Table 6).

### 4.7. Stepwise Regression Analysis of QOL among the Patients with FGIDs

The stepwise regression model for the QOL in patients with FGIDs included the depressive symptoms (*β* = −0.376, *p* < 0.001), resilience (*β* = 0.243, *p* < 0.001), social support (*β* = 0.120, *p* = 0.032), and childhood trauma (*β* = 0.99, *p* = 0.049), presenting an explanatory variance of 49.4% (F = 46.353, *p* < 0.001) for QOL (Table 7).

## 5. Discussion

In this study, we investigated the psychological characteristics and QOL of patients with upper and lower function GI disorders. We discovered that individuals with LGID exhibited higher depression and had lesser social support, lower resilience, and higher negative affectivity as compared to individuals with UGID, leading to lower QOL in individuals with LGID. 

FGIDs show a typical sex difference, with a higher prevalence in females than that male [34]. In addition, several socioeconomic factors of patients with FGIDs, such as marital status, educational level, and economic status were different from those of healthy controls [35]. However, studies about the differences in demographic characteristics between UGID and LGID have not been conducted. In this study, no differences were observed in the demographic factors between UGID and LGID groups. This result suggests that sex, age, and socioeconomic factors have limited association with the subtypes of FGIDs, and highlights the importance of understanding patient’s psychological characteristics.

In depression, higher K-BDI-II scores were observed in the LGID group than in the UGID group in its sub-domain, ‘emotional’. In general, depression is distinguished by cognitive-affective and somatic dimensions [36]. The cognitive–affective dimension includes the negative mood and negative affect, while the somatic dimension includes fatigue or loss of energy [37]. Pain was likely a major factor in the difference in emotional depression observed in this study. Emotional depression is an important factor in understanding pain as it causes individuals to continuously reflect on their physical challenges when faced with external stress or internal conflict [38]. In IBS patients, recurrent and chronic pain was observed [39]. On the contrary, in FH patients, pain was episodic, [40] and the FD (EPS) group showed improved symptoms after meal ingestion [22]. Therefore, it is possible that distress caused by pain has a more serious effect on patients with IBS than on patients with FD or FH. Thus, unlike cognitive and somatic depression, the difference in emotional depression may be attributed to the characteristic difference between UGID and LGID, i.e., UGID exhibits diverse symptoms, and in LGID, pain is a prerequisite symptom. In this study, since the effects of anxiety were relatively small among both UGID and LGID patients who visited the hospital, the difference between the groups was not significant. 

The patient’s personal history, such as physical, emotional, or sexual abuse, affects the severity of FGIDs, causing psychological distress and impacting daily function, eventually increasing healthcare seeking behavior [41]. In a study by Park et al., patients were divided into groups with IBS, FD, and FH for comparison of childhood trauma [42]. The findings of the study showed no significant difference in childhood trauma between the patient groups. Therefore, childhood trauma sems to be a characteristic seen widely in patients with FGIDs, which could be the reason for the absence of a significant difference between the UGID and the LGID groups. 

Social support is defined as the resources or interactions provided by others to help an individual cope with challenges [43]. In this study, the only significant difference between the UGID and LGID groups was observed in the sub-domain of ‘significant others’. Social support scale score was 32.05 for the UGID group and 29.14 for the LGID group, which is lower than the average score of 43.30, in a study by Shin et al. [29]. This suggests that both UGID and LGID groups have low social support compared to the normal control group, and that a low level of social support is considered a common psychological characteristic of people with FGIDs. 

Resilience is a dynamic process in which an individual who has undergone significant adversity or trauma demonstrates positive adaptive skills to cope with stress [44]. High resilience tends to be accompanied by psychological and physical health states, as well as a comprehensive ability to adequately adapt to various work and social situations [45]. In this study, significant differences were observed in the sub-domains of CD-RISC, hardiness and persistence, when comparing the UGID and LGID groups. In particular, the LGID group did not demonstrate significant differences in extrinsic factors such as optimism, support, and spirituality than the UGID group, although the LGID group demonstrated lower intrinsic factors such as hardiness and persistence. Such results demonstrate that psychotherapy or meditation techniques involving mindfulness to improve the internal resilience of LGID patients may be more effective options in treating LGID [46].

A type-D personality refers to a personality trait in which the individual engages in conscious oppression of self-expression in social interactions, while being vulnerable to negative emotions such as depression, anxiety, and stress [47]. Individuals with high NA often experience fear, anxiety, and excitement, while individuals with high levels of social restriction tend to be more tense and withdrawn when they are around others as a means to avoid fear [47,48]. A type-D personality is known as a determinant of psychological distress and is proposed as an important factor in predicting the prognosis of chronic diseases, particularly in coronary artery disease [49,50]. Although the LGID group demonstrated significantly higher NA scores than the UGID group, no difference in SI was observed between groups. Thus, although the tendency to suppress self-expression is similar across patients with UGID and LGID, those with LGID are more vulnerable to stress due to their higher NA, which eventually may have led to health-related fears, anxiety, and concern. Thus, support that involves empathy for patients with LGID is especially important.

QOL is a concept that integrates physical, emotional, and spiritual well-being. While depression, anxiety, and stress are known as factors that deteriorate QOL, social support and resilience are known as protective factors of QOL [51]. Patients with FGIDs report significantly lower QOL than the general population [14]. Here, the LGID group demonstrated significantly lower QOL than the UGID group, which may be because the abovementioned factors acted as protective or exacerbating factors, considering QOL involves various psychosocial and psychological factors. A step-by-step regression analysis performed to investigate the factors affecting the QOL of all patients with FGIDs identified depression (β = −0.376, *p* < 0.001), resilience (β = 0.243, *p* < 0.001), social support (β = 0.120, *p* = 0.032) and childhood trauma (β = 0.099, *p* = 0.049) as significant factors that accounted for 49.4% of the explanatory variance in overall QOL. Thus, for the improvement of QOL beyond the improvement of FGID symptoms, considering various psychosocial and psychological factors such as depression in the clinical setting is necessary.

This study had several limitations. First, the cross-sectional design of the study limited the examination of the causal relationship between each variable. Second, avoiding bias related to regional characteristics and the treatment environment was impossible as the study involved patients receiving care from one university hospital. Third, the mean age of the participants was high at 58 years, providing insufficient data on younger patients. Last, self-reported questionnaires were used to assess the participants’ psychological characteristics. Fourth, it would have been a better study if esophageal and gastric disorders were separated. Fifth, in the regression analysis for QOL, there is a possibility of high variance because variables and QOL had the same psychological dimension. Sixth, in the LGID group, only IBS was included, which limits the representation of LGID.

Nevertheless, the scientific design and clinical reliability of this study are excellent, as the study assessed psychological symptoms through a psychiatric interview on patients who were diagnosed with FGIDs. Furthermore, the strength of this study is that all the FGID patients were accurately diagnosed with FGID using endoscopy and GI physiological tests, and not just by using the ROME questionnaire. Furthermore, the results indicate that the effect of psychological factors is greater among patients with LGID than those with UGID. The results provide theoretical evidence for clinical case reports that have suggested the importance of psychological interventions such as selective serotonin reuptake inhibitors, psychotherapy, cognitive behavioral therapy in patients with IBS, compared to the standard treatment of UGID including FD using tricyclic antidepressants, which manages symptoms by acting on pain pathways.

## Figures and Tables

**Table 1 jcm-12-00124-t001:** Demographic and clinical characteristics of the participants.

Variables		UGID (*n* = 175)	LGID (*n* = 58)	*t*/*χ*^2^	*p*
Sex	Male	54(31.0)	16(27.6)	0.246	0.620
	Female	121(69.0)	42(72.4)		
Age		58.48 ± 13.10	58.62 ± 14.75	−0.069	0.945
Marital status	Unmarried	14(8.0)	4(6.9)	1.630	0.803
	Married	129(73.7)	41(70.7)		
	Separated	3(1.7)	1(1.7)		
	Divorced	12(6.9)	3(5.2)		
	Widowed	17(9.7)	9(15.5)		
Education (y)	None	5(2.9)	2(3.4)	6.347	0.175
	<7	31(17.7)	8(13.8)		
	7–9	34(9.4)	7(12.1)		
	10–12	64(36.6)	18(31.0)		
	>12	41(23.4)	23(39.7)		
Income (dollars/month)	<1000	60(34.3)	22(37.9)	2.214	0.819
	1000–2000	43(24.6)	15(25.9)		
	2000–3000	26(14.9)	7(12.1)		
	3000–4000	22(12.6)	4(6.9)		
	4000–5000	16(9.1)	6(10.3)		
	>5000	8(4.6)	4(6.9)		
Smoking	No	155(88.5)	53(91.4)	0.708	0.702
	Yes	20(11.5)	5(8.6)		
Alcohol	No	152(86.8)	49(84.5)	0.293	0.588
	Yes	23(13.2)	9(15.5)		
Anti-acidic agent	No	52(29.7)	18(31.1)	0.012	0.914
	Yes	123(70.3)	40(68.9)		
Chronic disease	No	91(52.0)	31(53.4)	0.037	0.848
	Yes	84(48.0)	27(46.6)		

Data are expressed as mean ± SD or *n* (%); UGID, Upper gastrointestinal disorders; LGID, Lower gastrointestinal disorders; M, mean; SD, standard deviation; *n*, number; Chronic disease: hypertension and diabetes mellitus.

**Table 2 jcm-12-00124-t002:** Comparison of depressive symptoms and anxiety between the UGID and LGID groups.

Variables		UGID (*n* = 175)	LGID (*n* = 58)	*t*	*p*
Depressive symptoms	Emotional	7.97 ± 5.49	10.64 ± 6.67	−3.031	0.003
	Cognitive	4.10 ± 4.10	5.24 ± 4.37	−1.801	0.073
	Somatic	6.46 ± 2.85	7.22 ± 3.12	−1.734	0.084
	Sum of depressive symptoms	18.53 ± 10.68	23.10 ± 12.68	−2.693	0.008
Anxiety	Reflecting subjective	4.29 ± 4.06	4.78 ± 4.68	−0.757	0.450
	Neurophysiological	4.53 ± 4.05	5.33 ± 4.34	−1.273	0.204
	Autonomic	2.01 ± 2.35	2.28 ± 2.56	−0.726	0.469
	Panic symptoms of anxiety	3.43 ± 2.49	3.33 ± 2.56	0.280	0.779
	Sum of anxiety	14.27 ± 11.28	15.71 ± 12.46	−0.819	0.414

Data are expressed as mean ± SD or *n* (%); UGID, Upper gastrointestinal disorders; LGID, Lower gastrointestinal disorders; M: Mean; SD, standard deviation; *n*, number.

**Table 3 jcm-12-00124-t003:** Comparison of childhood trauma and social support between the UGID and LGID groups.

Variables		UGID (*n* = 175)	LGID (*n* = 58)	*t*	*p*
Childhood trauma	Emotional neglect	16.43 ± 6.10	16.22 ± 5.61	0.232	0.817
	Physical abuse	6.37 ± 2.88	6.29 ± 2.53	0.171	0.864
	Sexual abuse	5.56 ± 1.72	5.84 ± 2.04	−1.040	0.299
	Emotional abuse	6.50 ± 2.98	6.69 ± 2.79	−0.433	0.666
	Physical neglect	11.29 ± 3.39	11.81 ± 2.77	−1.053	0.293
	Sum of childhood trauma	46.15 ± 9.68	46.86 ± 8.22	−0.504	0.615
Social support	Family	11.57 ± 3.72	11.07 ± 3.55	0.901	0.369
	Friends	9.25 ± 3.91	8.14 ± 3.67	1.907	0.058
	Significant others	11.23 ± 3.83	9.93 ± 3.70	2.254	0.025
	Sum of social support	32.05 ± 10.04	29.14 ± 9.24	1.953	0.052

Data are expressed as mean ± SD or *n* (%); UGID, Upper gastrointestinal disorders; LGID, Lower gastrointestinal disorders; M: Mean; SD, standard deviation; *n*, number.

**Table 4 jcm-12-00124-t004:** Comparison of resilience and type-D personality between the UGID and GID groups.

Variables		UGID (*n* = 175)	LGID (*n* = 58)	*t*	*p*
Resilience	Hardiness	20.72 ± 6.91	18.36 ± 6.83	2.259	0.025
	Persistence	19.89 ± 6.85	17.19 ± 7.40	2.546	0.012
	Optimism	9.49 ± 3.35	8.59 ± 3.92	1.709	0.089
	Support	5.11 ± 2.10	5.17 ± 1.88	−0.206	0.837
	Spiritual influence	3.98 ± 1.81	3.90 ± 1.77	0.316	0.752
	Sum of resilience	59.19 ± 17.78	53.21 ± 18.51	2.198	0.029
Type-D personality	Negative affectivity	11.08 ± 6.53	13.07 ± 6.71	−1.997	0.047
	Social inhibition	9.97 ± 7.07	11.66 ± 6.18	−1.624	0.106
	Sum of type-D personality	21.05 ± 12.84	24.72 ± 12.05	−1.919	0.056

Data are expressed as mean ± SD or *n* (%); UGID, Upper gastrointestinal disorders; LGID, Lower gastrointestinal disorders; M, Mean; SD, standard deviation; *n*, number.

**Table 5 jcm-12-00124-t005:** Comparison of QOL between the UGID and LGID groups.

Variables		UGID (*n* = 175)	UGID (*n* = 58)	*t*	*p*
QOL	Overall wellbeing	5.55 ± 1.50	4.74 ± 1.59	3.515	0.001
	Physical	20.15 ± 3.91	18.67 ± 4.12	2.471	0.014
	Psychological	18.61 ± 3.21	17.48 ± 3.77	2.207	0.028
	Social	8.50 ± 1.96	7.71 ± 2.17	2.613	0.010
	Environmental	25.96 ± 4.99	25.05 ± 5.13	1.192	0.235
	Sum of the QOL	78.78 ± 12.84	73.66 ± 13.19	2.615	0.010

Data are expressed as mean ± SD or *n* (%); UGID, Upper gastrointestinal disorders; LGID, Lower gastrointestinal disorders; M, Mean; SD: standard deviation; *n*, number; QOL, Quality of life.

**Table 6 jcm-12-00124-t006:** Bivariate associations between QOL and psychological variables (*n* = 233).

	QOL	Depressive Symptoms	Anxiety	Childhood Trauma	Social Support	Resilience	Type-D Personality
QOL	1						
Depressive symptoms	−0.610 **	1					
Anxiety	−0.473 **	0.657 **	1				
Childhood trauma	0.074	0.159 *	0.196 **	1			
Social support	0.411 **	−0.323 **	−0.247 **	0.202 **	1		
Resilience	0.488 **	−0.410 **	−0.235 **	0.172 **	0.495 **	1	
Type-D Personality	−0.491 **	0.533 **	0.448 **	0.129 *	−0.325 **	−0.405 **	1

* *p* < 0.05. ** *p* < 0.01; QOL, Quality of life.

**Table 7 jcm-12-00124-t007:** Stepwise regression analysis of QOL among the patients with FGIDs (*n* = 233).

	*β*	*t*	*p*	Adj R^2^	F	*p*
Depressive symptoms	−0.376	−6.155	<0.001	0.494	46.353	<0.001
Resilience	0.243	4.196	<0.001			
Social support	0.120	2.163	0.032			
Childhood trauma	0.099	1.976	0.049			

FGID: Functional gastrointestinal disorders; QOL, Quality of life.

## Data Availability

The datasets generated and/or analyzed during the current study are available from the corresponding author upon reasonable request.

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
