# Peer review of "Psychological Characteristics and Quality of Life of Patients with Upper and Lower Functional Gastrointestinal Disorders"

_jcm, 2022, doi:10.3390/jcm12010124_

Round 1

Author Response

Dear reviewers,

Thank you so much for your thoughtful review of the manuscript. We appreciate the work of you and the reviewers for the constructive comments and insightful critique of our manuscript. We fully addressed the comments and questions and have made revisions and clarifications in accordance with the reviewers’ comments. We believe that our manuscript is now much stronger and clearer. Below are the details of how we addressed the issues raised by the reviewers. We hope these revisions meet with your approval.

Introduction

  1. Introduction

: We revised introduction according to reviewer's opinion.

  1. Can the authors explain their specific research questions and a-priori hypotheses in the introduction?

 : We explained specific research questions and a-priori hypotheses in the introduction.

Materials and methods

  1. What was the classification criteria used to determine FH a part from true GERD/reflux hypersensitivity?

: We diagnosed FGIDs if patients complained of GI symptoms that satisfy Rome III criteria, but there was no evidence that organic disorders that are a cause of symptoms in various objective studies. In addition to diagnosis through symptoms questionnaires, this kind of complete objective diagnostic process could be a strength of this study. So we explained this in the method section of the revised manuscript as follows.

  1. What was the criteria the gastroenterologists used to determine globus, FH, FD, and IBS diagnosis?

: We used Rome III criteria because Korean-validated Rome IV questionnaire were not available at study period.

  1. What is the rationale for including the variables like Type-D personality?

: We added rationale for type-D personality in introduction.

  1. What was the rationale for the stepwise regression and how the authors chose the order to input the independent variables?

: In stepwise regression, the stepwise method of SPSS was used. The stepwise method is a mixture of forward selection and backward elimination. It starts with a model that does not contain any independent variables. After that, it enters the regression model in order from the variable with the lowest significance probability of the regression coefficient. Whenever a new variable is entered, the significance probability of all independent variables in the model is reevaluated, and if there is a variable with low significance, it is removed. And again, examine whether there are variables to include among the variables outside the model.

  1. Do the authors think theire regression analysis in circular?

: Agreed. It seems to be a weakness in research design. These problems were added to the limitation in discussion section.

  1. Do the authors have any questionnaire assessing symptom severity and disease duration?

: Thank you for the suggestion. It would be a very interesting analysis, but, unfortunately, we didn’t check symptom severity questionnaires such as IBS-SSS from participants in this study. This data will need to be obtained and analyzed in future studies.

  1. Where are the results/ what are the outcomes form the psychiatric interview?

: A mental status examination and evaluation of psychological characteristics through questionnaire were conducted by psychiatrist. Unfortunately, the results or outcoms of the psychiatric interview were not summarized. It is thought that research supplemented this will be needed in the future.

Results

  1. It may be helpful to label the measures in the methods section.

: We labeled measures in methods and tables.

  1. Are the authors powered to find effects given the differences in sample size between the UGID and LGID groups?

: It was the first study to assess a difference between UGID and LGID, so it was difficult to predict the sample size. In the future, it seems necessary to study the correction of the sample size between groups.

Discussion

  1. With the lack of data on pain severity/symptom severity, it fells too causal.

: Agreed. We revised the content according to the reviewer’s comments as followings.

  1. Catastrophizing is typically regarded as a cognitive construct closely related to anxiety.

: Agreed. We revised the content according to the reviewer’s comments as followings.

  1. Did the authors subtype their FD patients?

 : Agreed. The purpose of this study was to compare the upper GI and lower GI, so there was no comparison between FD subtypes. Compared to the lower IBS, where pain is an essential symptom, it is hypothesized that the upper part will be different from IBS because it is a disease group with various symptoms such as globus, early satiety, and fullness. In this study, it was found that the presence or absence of pain was related, and it is thought that a study is needed to compare EPS and PDS by gathering sufficient numbers of FD patients in the future.

  1. Much of the authors references are from those in the chronic pain.

: Agreed. We revised the content according to the reviewer’s comments in discussion.

  1. Catastrophizing is typically regarded as a cognitive construct closely related to anxiety.

: We revised the content according to the reviewer’s comments in discussion.

  1. The paper could be improved by removing the non-relevant literature and just focusing on what it means that there were not differences in anxiety between both groups.

 : We revised the content according to the reviewer’s comments in discussion.

  1. Why there were differences between upper and lower GI patients in social support.

: We added differences between upper and lower GI for social support in discussion.

  1. Catastrophizing is typically regarded as a cognitive construct closely related to anxiety.

: We revised the content according to the reviewer’s comments in discussion.

  1. I encourage the authors to really focus on the results and discuss clear hypotheses/discussion as to why this is the case.

 : We revised the content according to the reviewer’s comments in discussion.

Reviewer 2 Report

The present article presents an important and relevant study regarding the importance of treating and helping people with functional gastrointestinal disorders, and how important it is regarding the quality of life. Although the manuscript is well written, there are several major and minor corrections that should be implemented before its consideration for publication. The discussion section is very well written and comprehensive, but the references should also get some updates, as required in the introductory section also (see minor corrections).

Some major corrections:

-          The abstract should be more concise and according to journal guidelines: “The abstract should be a total of about 200 words maximum. The abstract should be a single paragraph and should follow the style of structured abstracts, but without headings...”

-          A very relevant update should be made on the whole manuscript as there are only 5 references from the last 5 years, and the majority are of more than 10 years old.

Some minor corrections:

-          the references are not formatted according to journal guideline

-          line 44 – 47 – this phrase should be based on more recent publications: 10.3389/fmed.2022.813204, 10.1136/postgradmedj-2017-135424

-          line 47 – 50 – recent references on altered microbiota: 10.3390/biology11040553

-          line 52 – the quality of life – should be abbreviated at first use in the text – as the abstract is considered a different text. Afterwards, the abbreviated form has to be used.

-          line 51 – 53 – this phrase needs a reference: 10.1016/j.neuron.2020.06.002

-          line 53 – the sentence is not finished

-          line 67 -68 -– this phrase needs a novel relevant reference: 10.1053/j.gastro.2020.10.066

-          line 74 – 76 – this section also needs a recent reference: 10.1016/j.jaim.2021.01.006

-          line 83 – the sentence should be finished

-          line 104 – the sentence is not finished

-          line 150 – the quality of life (QOL) – should not be abbreviated again, use the abbreviated form

-          table 1 – the UGID number from male and female gives the value of 174, and from income of 167. – revise the whole table for accuracy

If the authors implement all the required corrections, the manuscript can be published.

Author Response

Dear reviewers,

Thank you so much for your thoughtful review of the manuscript. We appreciate the work of you and the reviewers for the constructive comments and insightful critique of our manuscript. We fully addressed the comments and questions and have made revisions and clarifications in accordance with the reviewers’ comments. We believe that our manuscript is now much stronger and clearer. Below are the details of how we addressed the issues raised by the reviewers. We hope these revisions meet with your approval.

Some major corrections:

  1. The abstract should be more concise and according to journal guidelines: “The abstract should be a total of about 200 words maximum. The abstract should be a single paragraph and should follow the style of structured abstracts, but without headings...”

: We revised abstract according to reviewer's opinion.

  1. A very relevant update should be made on the whole manuscript as there are only 5 references from the last 5 years, and the majority are of more than 10 years old.

 : We revised abstract according to reviewer's opinion.

Some minor corrections:

  1. the references are not formatted according to journal guideline

: We revised references according to reviewer's opinion.

  1. line 44 – 47 – this phrase should be based on more recent publications: 10.3389/fmed.2022.813204, 10.1136/postgradmedj-2017-135424

: We changed the reference according to reviewer's opinion.

  1. line 47 – 50 – recent references on altered microbiota: 10.3390/biology11040553 (ref 2)

: We revised manuscript.

  1. line 52 – the quality of life – should be abbreviated at first use in the text – as the abstract is considered a different text. Afterwards, the abbreviated form has to be used.

: We revised manuscript according to reviewer's opinion.

  1. line 51 – 53 – this phrase needs a reference: 10.1016/j.neuron.2020.06.002

: We deleted this phrase.

  1. line 53 – the sentence is not finished

: We deleted this phrase.

  1. line 67 -68 -– this phrase needs a novel relevant reference: 10.1053/j.gastro.2020.10.066

: We deleted this phrase.

  1. line 74 – 76 – this section also needs a recent reference: 10.1016/j.jaim.2021.01.006

: We added reference according to reviewer's opinion.

  1. line 83 – the sentence should be finished

: We deleted this phrase.

  1. line 104 – the sentence is not finished

: We revised manuscript according to reviewer's opinion.

  1. line 150 – the quality of life (QOL) – should not be abbreviated again, use the abbreviated form

: We revised manuscript according to reviewer's opinion.

  1. table 1 – the UGID number from male and female gives the value of 174, and from income of 167. – revise the whole table for accuracy

 : We revised Table 1 according to reviewer's opinion.

Round 2

Reviewer 1 Report

Thank you to the authors for their edits to the manuscript. As mentioned in the first review, I appreciate the aims/design of the study and think the content is relevant. However, most of my concerns come with how the results are discussed, particularly that they are discussed in the context of other research that is either not relevant to the main findings or that they authors make causal claims without sufficient citations. Please see attached for my comments.

Author Response

Dear Reviewer,

Thank you so much for your thoughtful review of the manuscript. We appreciate the work of you and the reviewers for the constructive comments and insightful critique of our manuscript. We fully addressed the comments and questions and have made revisions and clarifications in accordance with the reviewers’ comments. We believe that our manuscript is now much stronger and clearer. Below are the details of how we addressed the issues raised by the reviewers. We hope these revisions meet with your approval.

Abstract

  • Authors use acronyms that are not previous defined (e.g., LGID/UGID)

: We revised abstract according to reviewer's opinion.

  • Authors did not define what the psychological questionnaires are measuring, so the results are confusing. For example, the sentence that says “LGID demonstrated higher scores in ‘emotional”…” the readers don’t know what “emotional” is because it’s not previously defined so it’s confusing. One suggestion is to use words that describe what the constructs are more broadly instead of using their subscale name.

: We revised abstract according to reviewer's opinion. [emotional -> emotional depression]

- Since the word limit for the abstract is 200 words, we were unable to include details of the questionnaire in it. We request your understanding on this issue.

  • Some grammatical errors where the authors added in new information, the authors should read over again.

: We revised grammatical errors according to reviewer's opinion.

Introduction

  • Line 46 - causal statement about psych “causing” GI symptoms. This should be toned down to not be so casual, instead I would suggest authors explain how psych can influence GI symptoms (e.g., peripheral and central/perceptual changes).

             : We revised manuscript according to reviewer's opinion.

  • Line 65 - how does social support lead to reduced stress levels?

:We revised manuscript according to reviewer's opinion.

“Social support appears to work by boosting the immune system, especially among people who are experiencing stress.”

Methods

  • The added information below the “diagnostic questionnaire of FGIDs” looks great, thank you for including this.
  • Is the lower GI disorder group just IBS? In that case, I think it can be misleading to call it “lower GI disorders” as the reader assumes it’s a diverse range of disorders like in the upper GI disorder group. However, instead it is just one disorder (IBS).

: Thank you for your comment. We agree with your opinion that further studies that include various FGIDs in the LGID group must be conducted. We have mentioned this in the limitations section of the discussion.

“In the LGID group, only IBS was included, which limits the representation of LGID.”

  • Did the authors look at associations between demographic factors and QOL? Age, for example, may be significantly associated with QOL and therefore would need to be controlled for in the regression.

: Thank you for your comment. We agree with your feedback. In this study, regression analysis was conducted using only psychological variables as demographic data could affect QOL.

Results

  • Can authors explain differences in more detail? For example, if there was a difference in “significant others” between groups, does that mean the UGID group experienced greater social support from significant others compared to the LGID?

:We revised manuscript according to reviewer's opinion.

“This suggests that the UGID group experienced greater social support from significant others compared to the LGID group.”

Discussion

  • Typo in ”famale” the second paragraph

: We revised grammatical errors according to reviewer's opinion.

  • Line 240 – FGIDs show a higher female prevalence compared to what group?

:We revised manuscript according to reviewer's opinion.

 “FGIDs shows a typical sex difference with higher female prevalence than male”

  • Line 247 – I’m not sure what the authors mean by “in consolidation, in the intervention for FGIDs”

 : We revised manuscripts according to reviewer's opinion.

  • Line 249- Authors state patients with UGID have higher depression scores than LGID, which is different than what is in the table.

: We revised manuscript according to reviewer's opinion.

  • Line 253 – FH and FD also experience visceral pain, it is a core component of FH. How does this hold up with the author’s theory about pain being the differential factor

: We revised manuscript according to reviewer's opinion.

“In IBS patients, recurrent and chronic pain was observed. On the contrary, in FH patients, pain was episodic, and the FD (EPS) group showed improved symptoms after meal ingestion. Therefore, it is possible that distress caused by pain has a more serious effect on patients with IBS than on patients with FD or FH.”

  • Line 258 – stating that patients become “obsessed with pain” can be viewed as stigmatizing, I would encourage the authors to use different language to avoid further stigmatizing an already stigmatized group.

: We deleted sentence according to reviewer’s opinion

  • Paragraph starting with line 262 – I don’t understand the relevance of this paragraph, how does a study comparing differences between healthy controls and patients with FGIDs who did/did not visit a hospital has to do with the current study. This seems irrelevant and confusing. Wouldn’t the authors just argue that anxiety did not differ between the groups because anxiety is relevant in both classes of disorders?

: We deleted sentences according to reviewer’s opinion

  • Same thing for the next paragraph, this seems to be missing the main outcome of the results which was that there was not a difference in Childhood abuse history between Upper and lower GI disorders – why compare it to other somatic disorders? Just explain the results and discuss why this may be the case (e.g., discussing literature that shows abuse history is prevalent in patients with IBS and in other FGIDs, thus that’s likely why there were not differences)

: We revised manuscript according to reviewer's opinion.

“In a study by Park et al., patients were divided into groups with IBS, FD, and FH for comparison of childhood trauma. The findings of the study showed no significant difference in childhood trauma between the patient groups. Therefore, childhood trauma sems to be a characteristic seen widely in patients with FGIDs, which could be the reason for the absence of a significant difference between the UGID and the LGID groups.”

  • The social support paragraph – there lacks citations for some of the statements made (e.g., LGID pts use the bathroom more which negatively affects their social life – how do you know this is true? Need to provide a citation). Next, the paper you’re citing alleging that it’s more difficult to communicate to others when you have an LGID compared to an UGID did not compared lower to upper, it compared IBS to IBD, so this is an inaccurate representation of the article. The authors can provide hypotheses for why they feel like LGID have increased social support, but making statements without/with inaccurate citations is not good practice. Additionally, have the authors considered what the reverse means? Why do UGIDs report lower support?

: We revised manuscript according to reviewer's opinion.

“ Social support scale score was 32.05 for the UGID group and 29.14 for the LGID group, which is lower than the average score of 43.30, in a study by Shin et al. This suggests that both UGID and LGID groups have low social support compared to the normal control group, and that a low level of social support is considered a common psychological characteristic of people with FGIDs.”

  • Line 288 – not all patients develop psychological symptoms, so authors cannot make the statement that patients will eventually experience psych symptoms.

: We deleted sentence according to reviewer’s opinion.

  • Line 297 – what is the evidence for the statement that psychotherapy/mediation improve resilience (e.g., no citation)

: We added references according to reviewer’s opinion.

“Joyce, S.; Shand, F.; Tighe, J.; Laurent, S.J.; Bryant, R.A.; Harvey, S.B. Road to resilience: a systematic review and meta-analysis of resilience training programmes and interventions. BMJ. Open. 2018, 8(6), e017858.”

  • Line 308 – Need citation for statement that higher NA leads to more stress in people in general/those with DGBIs

: We added references according to reviewer’s opinion.

“ Muscatello, M.R.; Bruno, A.; Mento, C.; Pandolfo, G.; Zoccali, R.A. Personality traits and emotional patterns in irritable bowel syndrome. World J Gastroenterol 2016, 22(28), 6402-15.”

Reviewer 2 Report

The manuscript has been considerably improved. However, there are still some terms that have to be better defined, and the authors should pay more attention to the correctness of all the used terms and abbreviations.

Some  concrete corrections:

- in the abstract line 15 Qol was first abbreviated in uppercase. Please revise the whole manuscript as there are several other similar mistakes.

- line 16 - questionnairess > correct to questionnaires

- line 16 - please first indicate the abbreviation LGID - in line 15 (lower gastrointestinal disorder group) - the same applies for UGID also

line 24 - OQL>QOL

- please revise the formatting of references according to the author's guideline

- line 40 - The term dysbiosis is a simplification and can be used as a mental shortcut. The authors should make readers aware of gut microbiota's individual and complex nature. It is important to emphasise that there is no "healthy" microbiota > as can be seen: 10.3389/fmed.2022.813204; 10.3390/microorganisms10040763 

- line 50 - 51 - this phrase needs references! (10.3390/ijerph19031208, https://www.ncbi.nlm.nih.gov/pmc/articles/PMC6469458/)

- line 73 - please define the abbreviation FD

- line 91 - how about FH?

- line 252 - please correct affect > effect

Author Response

Dear Reviewer,

Thank you so much for your thoughtful review of the manuscript. We appreciate the work of you and the reviewers for the constructive comments and insightful critique of our manuscript. We fully addressed the comments and questions and have made revisions and clarifications in accordance with the reviewers’ comments. We believe that our manuscript is now much stronger and clearer. Below are the details of how we addressed the issues raised by the reviewers. We hope these revisions meet with your approval.

Some  concrete corrections:

- in the abstract line 15 Qol was first abbreviated in uppercase. Please revise the whole manuscript as there are several other similar mistakes.

: We revised abstract according to reviewer's opinion.

- line 16 - questionnairess > correct to questionnaires

: We revised abstract according to reviewer's opinion.

- line 16 - please first indicate the abbreviation LGID - in line 15 (lower gastrointestinal disorder group) - the same applies for UGID also

: We revised abstract according to reviewer's opinion.

- line 24 - OQL>QOL

: We revised abstract according to reviewer's opinion.

- please revise the formatting of references according to the author's guideline

: We revised references according to reviewer's opinion.

- line 40 - The term dysbiosis is a simplification and can be used as a mental shortcut. The authors should make readers aware of gut microbiota's individual and complex nature. It is important to emphasise that there is no "healthy" microbiota > as can be seen: 10.3389/fmed.2022.813204; 10.3390/microorganisms10040763 

:: We revised references according to reviewer's opinion.

- line 50 - 51 - this phrase needs references!

 (10.3390/ijerph19031208, https://www.ncbi.nlm.nih.gov/pmc/articles/PMC6469458/)

: We added reference according to reviewer's opinion.

- line 73 - please define the abbreviation FD

: We defined the abbreviation FD according to reviewer's opinion.

- line 91 - how about FH?

: We defined the abbreviation FH according to reviewer's opinion.

- line 252 - please correct affect > effect

: We think negative affect is correct in line 252.